# Identifying the Multitarget Pharmacological Mechanism of Action of Genistein on Lung Cancer by Integrating Network Pharmacology and Molecular Dynamic Simulation

**DOI:** 10.3390/molecules29091913

**Published:** 2024-04-23

**Authors:** Raju Das, Joohan Woo

**Affiliations:** 1Department of Physiology, College of Medicine,, Dongguk University Wise, Gyeongju 38066, Republic of Korea; rajudasbgc@gmail.com; 2Channelopathy Research Center (CRC), College of Medicine, Dongguk University Wise, 32 Dongguk-ro, Ilsan Dong-gu, Goyang 10326, Republic of Korea; 3Medical Cannabis Research Center, College of Medicine, Dongguk University Wise, 32 Dongguk-ro, Ilsan Dong-gu, Goyang 10326, Republic of Korea

**Keywords:** genistein, lung cancer, AKT1, EGFR, network pharmacology, molecular dynamics simulation

## Abstract

Food supplements have become beneficial as adjuvant therapies for many chronic disorders, including cancer. Genistein, a natural isoflavone enriched in soybeans, has gained potential interest as an anticancer agent for various cancers, primarily by modulating apoptosis, the cell cycle, and angiogenesis and inhibiting metastasis. However, in lung cancer, the exact impact and mechanism of action of genistein still require clarification. To provide more insight into the mechanism of action of genistein, network pharmacology was employed to identify the key targets and their roles in lung cancer pathogenesis. Based on the degree score, the hub genes AKT1, CASP3, EGFR, STAT3, ESR1, SRC, PTGS2, MMP9, PRAG, and AR were significantly correlated with genistein treatment. AKT1, EGFR, and STAT3 were enriched in the non-small cell lung cancer (NSCLC) pathway according to Kyoto Encyclopedia of Genes and Genomes analysis, indicating a significant connection to lung cancer development. Moreover, the binding affinity of genistein to NSCLC target proteins was further verified by molecular docking and molecular dynamics simulations. Genistein exhibited potential binding to AKT1, which is involved in apoptosis, cell migration, and metastasis, thus holding promise for modulating AKT1 function. Therefore, this study aimed to investigate the mechanism of action of genistein and its therapeutic potential for the treatment of NSCLC.

## 1. Introduction

Food-derived bioactive compounds are well known for their multifactorial properties and functions, possessing preventive action against poor cognition, cardiovascular disease, obesity, hypertension, and cancer. They have gained potential therapeutic attention for cancer prevention, thus holding therapeutic concern as a promising source of cancer mimetics [1,2,3,4,5]. Genistein, a natural isoflavone derived from *Genista tinctoria* L., is popular because of its estrogen-like structure along with promising antioxidant, anti-inflammatory, antibacterial, and antiviral effects; however, it has been shown to have anticancer effects in recent years and is associated with apoptosis and metastasis modulators through several signaling cascades [6,7,8]. Evidence suggests that genistein triggers apoptosis through the over-activation of caspase-9 and caspase-3 in Hela cervical cancer cells and p38 in the colon cell line HT-29 via the MAPK signaling pathway [9,10]. In addition, genistein regulates antiapoptotic proteins, such as Bcl-2 and coulBax (proapoptotic proteins), which induce apoptotic cell death in LoVo and HT-29 cancer cell lines [7,11]. Genistein exerts an antimetastatic effect in the A549 lung cancer cell line by regulating the MMP-2 and PI3K/Akt pathways while playing a role in gastric cancer by inhibiting ERK1/2 and angiogenesis in ovarian cancer cells and reducing the protein levels of vascular endothelial growth factor, a potential target of ovarian cancer treatment [12,13].

Moreover, genistein has been reported to alter FoxM1 (forkhead box protein M1) function, which triggers apoptosis and cell cycle arrest in the H446 small cell lung cancer cell line, halting cell proliferation and differentiation [14]. The combination of genistein with other compounds is effective against non-small cell lung cancer (NSCLC) cell lines by targeting mutated EGFR, showing a synergistic anticancer effect; however, the exact role of genistein in lung cancer remains unclear [15]. Focusing on the fundamental role of genistein in lung cancer, we designed the present study based on network pharmacology. We applied molecular docking and molecular dynamics (MD) simulations to unravel the interaction of genistein with its potential targets, which may affect several cancer-related pathways. 

Lung cancer (LC) is categorized into SCLC and NSCLC; however, most patients with LC have NSCLC with metastasis at diagnosis [16]. A series of phenomena, such as long-term immune imbalance, gene defects, epigenetic changes, and oxidative stress, are standard pathological features that alter the balance of molecular mechanisms, leading to oncogene activation and tumor-suppressor gene inactivation. This manifests as changes in different cellular events, such as cell growth and proliferation, metabolism, angiogenesis, and apoptosis [17,18,19,20,21,22]. Therefore, identifying the signaling pathways and their related target genes that drive these events is pivotal for determining the exact phenomena and developing treatment strategies. The current management of lung cancer involves surgical intervention, radiotherapy, and chemotherapy, all of which have extreme side effects. Targeted therapy has become popular because of its minimal side effects [23,24]. 

Network pharmacology makes it easier to understand the interaction between a particular drug and its target proteins, which helps to identify protein-enriched signaling pathways highly associated with the disease paradigm [25]. Therefore, we aimed to identify therapeutic target proteins in lung cancer, particularly in NSCLC, and unravel the impact of genistein on these target proteins at the molecular and cellular levels. Based on the highest degree score, the hub genes STAT3, EGFR, PPARG, CASP3, PTGS2, MMP9, AR, ESR, and AKT1 were highly enriched in the cancer pathway according to the Kyoto Encyclopedia of Genes and Genomes (KEGG) pathway analysis, suggesting direct or indirect involvement in lung cancer progression. Moreover, molecular docking and MD simulations were sequentially performed to predict the binding affinity of genistein to crucial hub genes. Genistein showed significant binding affinity to AKT1, thus raising its therapeutic potential for treating NSCLC by targeting AKT1.

## 2. Results

### 2.1. Identification of Target Genes

The construction of weighted genes related to lung cancer is a potential therapeutic intervention. Two disease-associated gene databases—GeneCards and DisGeNET—were used to construct a comprehensive network of lung cancer target genes. In total, 1054 genes associated with lung cancer, with a relevance score ≥ 20, were obtained from the GeneCards database, while 4081 target genes were downloaded from the DisGeNET database. Subsequently, the targets obtained from the two databases were merged through the elimination of overlapping targets, and a total of 4285 target genes were identified for inclusion in the next steps. Meanwhile, 254 promising target genes related to genistein were identified using the SwissTargetPrediction, Binding DB, PharmMapper, and STITCH databases. Subsequently, a total of 4285 disease-related targets and 254 compound-related targets were found to intersect using VENNY 2.1 to obtain overlapping targets, as shown in Figure 1 [26]. These represent the final 129 common targets.

### 2.2. Protein–Protein Interaction Network Construction and Hub Gene Prediction

Proteins facilitate biological functions in a cell independently or together by interacting with other proteins. Thus, identifying protein–protein interactions is a potential approach to understanding protein function in a biological system. To determine the critical targets and their interactions corresponding to lung cancer pathogenesis, 129 targets were subjected to network analysis in the STRING database, and the network was introduced into Cystoscape 3.10.1 [27]. 

The network consisted of 128 nodes and 2434 edges, with a local clustering coefficient of 0.587, indicating the total number of targets connected to the network. In addition, the scores for degree centrality (DC), betweenness centrality, and closeness centrality also signified the properties of each node. The details of these parameters are shown in the Appendix A. Among the 129 targets, the top 10 core targets were further screened using the CytoHubba Plugin of Cytoscape based on the DC score, where dark red nodes indicated high susceptibility to lung cancer pathogenesis. According to the highest degree of centrality score, the dominant hub targets were AKT1, CASP3, EGFR, STAT3, ESR1, SRC, PTGS2, MMP9, PPARG, and AR (Figure 2B and Appendix A).

### 2.3. Gene Interaction Analysis Using GeneMANIA

GeneMANIA emphasizes the co-expression, physical interactions, and colocalization of genes with similar functions based on genomic and proteomic data. The network generated by GeneMANIA demonstrated a complex functional network of hub targets and their related genes, where the dark pink nodes represented the hub targets and the green nodes were recognized as related genes. The network possessed 30 nodes, 268 edges, a clustering coefficient value of 0.561, a co-expression level of 43.02%, 32.80% physical interactions, 6.82% colocalization, and 4.85% shared protein domain levels. The results are shown in Figure 3.

### 2.4. Functional Gene Ontology and KEGG Pathways

The role of hub targets in biological systems was clarified by gene ontology (GO) enrichment, and KEGG analyses conducted using Shiny GO 0.77. The top 20 results in terms of GO biological process (BP), cellular component (CP), and molecular function (MF) categories and KEGG pathways are shown in Figure 4. The BP category was highly enriched in the cellular response to organic cyclic compounds, reproductive process, response to oxygen-containing compounds, cellular response to endogenous stimulus, inflammatory response, regulation of the apoptotic process, and regulation of programmed cell death, whereas the CP category was enriched in membrane rafts, membrane microdomains, synapses, chromatin, and organelle envelopes. In addition, MF terms were highly involved in molecular transducer activity, signaling receptor activity, kinase binding, protein kinase binding, and transcription factor binding. The top 20 signaling pathways were then investigated using KEGG enrichment analysis. The cancer pathway was highly enriched with the highest number of hub genes, suggesting the direct or indirect involvement of these genes in cancer progression.

### 2.5. Analysis of the NSCLC-Related Pathways and Key Protein Targets

We focused on pathways and hub targets involved in NSCLC progression. According to the KEGG pathway analysis, EGFR, AKT1, and STAT3 were enriched in NSCLC progression. EGFR belongs to the ErbB family, and its expression is regulated via a specific transcription factor. Overexpression of EGFR due to mutations and its activation upon ligand binding to particular domains trigger downstream pro-oncogenic signaling pathways, leading to cell proliferation and antiapoptotic effects.

AKT is a serine/threonine kinase and an essential component of the PI3K-AKT signaling pathway. The PI3K-AKT signaling pathway is activated by several growth factors, and activated AKT promotes different cellular events, such as resistance to apoptosis, cell proliferation, survival, migration, and protein expression, but also accelerates metastasis. In general, AKT exists in three isoforms: AKT1, AKT2, and AKT3. The AKT1 isoform is associated with NSCLC and is a metastasis modulator [28]. In addition, activation of STAT3 (signal transducer and activator of transcription 3) has been reported to promote NSCLC development, which can be stimulated by either receptor tyrosine kinases (EGFR, VEGFR) or non-receptor tyrosine kinases (JAKs, c-SRC), leading to angiogenesis and cell survival events [29]. More importantly, EGFR, AKT1, and STAT3 are well-known biomarkers of cancer, but the effects of genistein on these targets remain elusive. The genistein-integrated NSCLC pathway is shown in Figure 5.

### 2.6. Molecular Docking Analysis

The ability of genistein to bind to the target protein of interest was verified using molecular docking and MD simulations. Redocking was performed to predict the correct binding pose. The RMSD difference between the docked pose and crystal structure was 0.030 Å. After that, we employed extra-precision (XP) docking to predict genistein interactions and selected several known positive controls for comparison with genistein-bound EGFR, AKT1, and STAT3 complexes. Binding affinities in terms of docking and free energies are listed in Appendix A. According to docking and molecular mechanics with generalized Born and surface area solvation (MM-GBSA) scores, genistein showed a moderate affinity with EGFR and the lowest affinity with STAT3 but the highest affinity with AKT compared with other positive controls. Thus, genistein bound to and modulated AKT1 function. We considered individual three times 200 ns MD simulation of the genistein, MK-2206, and H-89-bound AKT1 complex for further verification.

The docking pose of AKT1 lies in a two-dimensional interaction, which represents the types of interactions between the target protein and the reference ligand and is presented in detail in Appendix A. AKT1 is an isoform of AKT that consists of an N-terminal pleckstrin homology (PH) domain, an interdomain linker, a kinase domain, and a C-terminal hydrophobic motif or regulatory domain. As shown in a previous study, the interaction between the PH and the kinase domain changes AKT1 from an active to an inactive state under basal conditions. 

In contrast, the PH domain and phosphoinositide initiate phosphorylation of Thr 308, which triggers full activation of the interaction of AKT1 with phosphorylated Ser 473, is part of the hydrophobic motif of the C-terminus. The majority of existing Akt inhibitors target ATP-binding sites but are non-selective blockers because they share 50% identical residues. Therefore, targeting the allosteric site is an alternative method for designing new Akt inhibitors. The AKT1-bound inhibitor revealed that both the PH and kinase domains participate in allosteric inhibition, for which interactions with Trp 80, Ser 205, and Lys 268 are essential. Figure 6A shows the residue-wise structure of the AKT1 domain and superimposed docked complex. Genistein and AKT1 demonstrated hydrogen and hydrophobic interactions as investigated using a three-dimensional (3D) interaction pattern, where Gln 79 made both hydrogen and hydrophobic contact, while Leu 210 and Leu 264 exhibited hydrophobic interactions with genistein. Genistein also made direct hydrogen contact with Ser 205, as evident in the AKT1-inhibitor VIII complex [30]. The selective inhibitor MK-2206 showed several hydrophobic interactions with Trp 80, Leu 210, Leu 264, Val 270, Tyr 272, and Asp 274 and hydrogen bonds with Asn 53 and Tyr 27. The interaction between H-89 and AKT1 differed from the interaction between genistein and MK-2206 by showing hydrogen bond hydrophobic and π-cation interactions. Residues Asn 53, Gln 79, Trp 80, Leu 210, and Val 270 made hydrophobic interactions, while Trp 80 made only a π-cation interaction with H-89. Only Gln 79 made a hydrogen bond with H-89. The 3D interactions of genistein, MK-22006, and H-89 with AKT1 are shown in Figure 6B.

### 2.7. MD Simulation

The docking results were verified using time-dependent MD simulations, providing significant structural insights, such as stability and conformational changes, based on ligand binding to the target protein. Two hundred nanosecond MD simulations corresponding to each complex were considered, and the trajectories derived from the simulation were analyzed to determine the level of structural changes by calculating the root mean square deviation (RMSD) of the protein and ligand. The total intermolecular protein–ligand contact during MD simulation also represented the critical residues responsible for complex stability or conformational changes, which were also considered for analysis.

#### 2.7.1. Protein–Ligand RMSD

RMSD is a parameter used to determine dynamical stability measured by the displacement of the alpha carbon (Cα) with time. Figure 7 shows the total RMSD calculations for all the docked complexes. For genistein, AKT1 displayed a preliminary fluctuation to reach system equilibration (Figure 7B). After 50 ns of each run, AKT1 remained stable until the end of the simulation period without showing any major fluctuations. The RMSD changed from 2.0 to 2.5 Å during this period, suggesting structural stability. Similar to the protein RMSD value, genistein remained stable after approximately 65 ns and maintained this trend until 200 ns in run 1. The protein RMSD result of run 2 also showed almost identical results, while run 3 showed a slightly higher RMSD value compared with run 1. Therefore, genistein made less structural changes to AKT1 based on overall three individual simulation runs (Figure 7B).

In contrast, MK-2206-bound AKT1 displayed overall stability throughout the simulation period in each run. Similar results were observed for MK-2206, which showed no fluctuation, indicating that MK-2206 was attached to the AKT1 binding site during the simulation (Figure 7C and Figure 8B). In addition, the protein RMSD value of AKT1 induced by H-89 showed several fluctuations; however, it remained stable after 150 ns and until the end of the simulation (Figure 7D and Figure 8C). 

In addition, the average RMSD value of H-89 was higher than that of the genistein- and MK-2206-bound structures. Based on the ligand RMSD value of H-89, it was evident that H-89 maintained stability from the initial simulation time to 150 ns but decreased to 1.5 Å and again attained equilibrium until the end of the simulation time in run 1. Figure 8 shows the ligand RMSD values of genistein, Mk-2206, and H-89. Therefore, based on the overall protein–ligand RMSD values, genistein and MK-2206 demonstrated the possibility of more stable binding compared with H-89. Overall, protein–ligand RMSD suggested that genistein showed an identical result to MK-2206 without any significant fluctuation, and showed better stability compared with H-89. 

#### 2.7.2. Total Protein–Ligand Contact Profiling

The functional residues bound to the allosteric site of AKT1 were verified by total protein–ligand contact during the simulation. Interactions involving >30% of the simulation time were considered. To provide further insights, the entire interaction pattern is shown in Figure 9, Figure 10 and Figure 11. We observed that the total protein–ligand contact was confined to hydrogen bond hydrophobic and water bridge interactions with conserved and non-conserved residues. As shown in Figure 9, Figure 10 and Figure 11, all complexes showed hydrophobic interactions with the conserved residue Trp 80, which was reportedly vital for holding the ligand to the allosteric site. 

In addition, the interaction similarity was favored by Gln 79, Leu 264, Lys 268, and Tyr 272, where Lys 268 supported the same evidence regarding the AKT1 inhibition hypothesis in the allosteric site, which was also recognized by genistein in the same way. Genistein formed hydrophobic interactions with the conserved Trp 80 residue and hydrogen bonds with the Lys 268 residue (Figure 9). However, MK-2206 required interaction with another principal residue, Tyr 272, along with Trp 80 to inhibit AKT1, which was apparently seen throughout the simulation period, thus aligning with previous findings (Figure 10). Asn 204, Ser 205, Leu 264, Glu 267, Lys 268, Asp 274, and Asp 292 also made contact with Mk-2206 (Figure 10). Unlike genistein and MK-2206, H-89 had the highest level of contact only with Trp 80, but had less interactions with other residues, such as Gln 79, Leu 210, Leu 264, Val 270, Val 271, Tyr 272, and Asp 292, which may explain several fluctuations in the protein RMSD values (Figure 7D). Hydrogen bonds, hydrophobic interactions, and water bridges are represented by green, light purple, and blue color (Figure 9, Figure 10 and Figure 11).

According to Figure 9B,C, Trp 80, Lys 268, and Tyr 272 were highly contributed in genistein binding, which supported the allosteric inhibition of AKT1 by genistein. In the case of MK-2206, a similar result was also observed in runs 2 and 3 (Figure 10B,C). In contrast, H-89 bound AKT1 was not stable due to lack of less hydrogen bonds seen in total protein–ligand contacts in run 2 and run 3 (Figure 11B,C). 

The contribution of residual interactions per trajectory was further validated using a time-dependent protein–ligand plot, depicted in the Appendix A, representing the number of contacts and their density. More than one contact in each trajectory was shaded dark. In the AKT1-genistein complex, residue Trp 80 interacted with genistein throughout the simulation time and residues Asn 53, Lys 268, and Ile 290 were involved in ligand binding from approximately 60 ns to the end of the simulation period through the water bridge and hydrogen and hydrophobic contacts, which supported stable protein RMSD values.

In general, Trp 80 and Lys 268 were evolutionarily conserved, suggesting their crucial role in genistein binding to the allosteric site of AKT1. In the AKT1-MK2206 complex, MK2206 showed consistent stability in the AKT1 pocket throughout the simulation period, interacting with residues Trp 80, Asn 204, Tyr 272, and Asp 274 via hydrophobic, hydrogen, and water bridge contacts, resulting in a stable protein–ligand RMSD value. In addition, the participation of residues Leu 264 and Glu 267 contributed to MK2206 stability, which was assumed after 70 ns and extended to the end of the simulation time; thus, it may have contributed to protein–ligand stability. In contrast, H-89 remained stable and was retained only by the conserved Trp 80, primarily through hydrophobic contacts.

## 3. Discussion

A series of characteristics have made NSCLC a dominant subtype of lung cancer. Among them, metastasis is central to NSCLC pathophysiology and is the main reason for treatment failure and high mortality rates worldwide [31,32]. Evidence suggests that most patients diagnosed with metastases are resistant to chemotherapy. The exact underlying mechanism of metastasis is complex and still unclear; therefore, identifying novel therapeutic targets and their potential modulators is another option for developing NSCLC treatments [16]. Thus, food supplements may be a more significant source of therapeutic agents than synthetic compounds. Polyphenolic phytochemicals are bioactive compounds that are mostly isolated from plants and possess defense mechanisms against stress and microbial attacks. In addition to their diverse biological activities and beneficial outcomes, they are well known for their antimutagenic and anticarcinogenic properties against several types of cancer. The consumption of genistein-rich soy protein reduces metastasis, as reported in several preclinical studies; however, the exact mechanisms are still unknown. Our analysis identified the top hub genes that were highly correlated with genistein. Among them, STAT3, EGFR, PPARG, CASP3, PTGS2, MMP9, AR, ESR1, and AKT1 were enriched in the cancer pathway according to the KEGG pathway analysis, suggesting the anticancer action of genistein.

Caspase-3 (CASP3) is a cysteine protease that causes DNA fragmentation and induces apoptosis [33]. The activation of CASP3 induces radiotherapy sensitivity and reduces H460 lung cancer cell survival [34]. In addition to increasing sensitivity to chemotherapy and radiotherapy, CASP3 is involved in the inhibition of metastasis and cancer cell invasion [35]. In contrast, excessive production of prostaglandin-endoperoxide synthase-2 (PTGS2) leads to lung cancer progression, which subsequently leads to metastasis and angiogenesis [36]. PTGS2 converts arachidonic acid into prostaglandins during inflammation. Multiple studies have indicated that PTGS2 levels are increased in lung tumor cells [37]. Matrix metalloproteinase (MMP)-9 is a member of the MMP family and regulates tissue remodeling. In general, during inflammation, inflammatory molecules and hormones trigger the elevation of MMP levels in affected tissues to promote tissue repair. The reduction in MMP-9 levels reduces lung cancer metastasis, while Li et al. showed that the leaves of *Punica granatum* inhibit cell invasion and migration due to the suppression of MMP-2 and MMP-9 levels [38]. Although estrogen receptor (ESR1 or ERα protein) involvement in NSCLC progression is still controversial, ESR1 modulates NSCLC development and progression, triggering EGFR as well as the Wnt/β-catenin pathways. EGFR activation triggers downstream signals and their effector molecules, ultimately leading to cell survival, proliferation, and differentiation [39].

Furthermore, peroxisome proliferator-activated receptors (PPARs) play a significant role in malignancies. PPARG activation terminates tumor growth and metastasis through anti-inflammatory and antiangiogenic cascades [40]. Thus, given that the identified hub genes were highly associated with the onset of lung cancer, the correlation between genistein and these hub genes suggests that genistein may function as a potential modulator. In addition, CASP3, PTGS2, MMP9, and AKT1 are associated with the TNF signaling pathway, which is a key signaling pathway susceptible to the inflammatory response initiated by NF-κB activation [41,42,43]. The inflammatory response becomes chronic in cancer and can result in mutation and cell proliferation, creating an environment conducive to cancer development [44]. 

This study focused on the pathways and their associated target genes involved in NSCLC development. KEGG signaling pathway analysis showed that AKT1, EGFR, and STAT3 were enriched in the NSCLC pathway, indicating that altering the functions of these genes can enhance NSCLC progression. AKT is also known as protein kinase B and it participates in several biological cascades in response to stimuli. Its involvement in cell proliferation, cell survival, and apoptosis means that it is a potential biomarker in several types of cancer, such as lung cancer. In general, AKT activation accelerates cell growth via the PI3K-AKT pathway, resulting in the phosphorylation of several proteins. AKT1 is an isoform of AKT that possesses an anti-lung-cancer effect [45]. Sequential ablation of AKT1 in tobacco- and K-Ras-treated mice inhibits further lung tumor progression, and the same phenomenon has been observed in IGF-IR positive lung cancer [45,46]. Moreover, treatment with a selective inhibitor of AKT1 initiates apoptosis [47]. Additionally, the involvement of AKT1 in cell migration and invasion highlights its importance in metastasis [47]. EGFR is a transmembrane receptor tyrosine kinase involved in epithelial tissue development. However, mutations in EGFR enhance antiapoptotic signaling and cause excess cell proliferation and may result in early metastasis in lung and breast cancer patients [48]. The mammalian PI3K/AKT pathway is considered one of the major intracellular pathways that is frequently altered in cancers and drives several uncontrolled cellular events. In addition, activation of the PI3K/AKT pathway is evident in EGFR-TKI resistance events [49,50]. Thus, targeting components of the PI3K/AKT pathway and their potential modulators may halt or prolong metastasis. Therefore, we determined the binding affinity of genistein to EGFR and AKT1 using molecular docking and MD simulations. Based on the docking results, genistein showed a significant binding affinity for AKT1 and EGFR compared with STAT3. Genistein-bound AKT1 was more stable than MK-2206- and H-89-bound AKT1, according to the MD simulation results, indicating its potential inhibitory activity against AKT1 [51,52]. Overall, genistein mediated anticancer pathway is postulated in Figure 12.

## 4. Materials and Methods

### 4.1. Screening of Lung Cancer Targets

GeneCards and Disgenenet were used to retrieve target genes related to lung cancer [53,54]. The species was limited to “human”, and the keywords used in the search were “lung cancer”. Lung-cancer-associated target genes were compiled and merged. Subsequently, the identified genes were sorted and verified using UniProt based on the protein and host names [55].

### 4.2. Screening of Genistein-Related Targets

Genistein-related targets were acquired using SwissTargetPrediction (http://www.swisstargetprediction.ch/, accession date: 27 January 2023), Binding DB (https://www.bindingdb.org/rwd/bind/index.jsp, accession date: 27 January 2023), Pharm Mapper (https://www.liab-ecust.cn/pharmmapper/, accession date: 20 February 2023), and STITCH (http://stitch.embl.de, accession date: 27 January 2023) [56,57,58,59]. The canonical SMILES and 3D structure of genistein were used to identify related genes. Subsequently, the acquired targets were validated using the UniProt database and sorted by eliminating redundant targets.

### 4.3. GeneMANIA Analysis

To investigate functionally similar query genes, the GeneMANIA plugin of Cytoscape software (Version-3.10.1) was utilized, using Homo sapiens as a reference. GeneMANIA predicts the following genetic pathways: shared protein domains, colocalization, physical interactions, and co-expression levels of gene sets [60]. The top hub genes were subjected to analysis using the GeneMANIA plugin of Cytoscape software. 

### 4.4. Molecular Docking

#### 4.4.1. Ligand Preparation

Appropriate ligands were first retrieved from the PubChem database [61] and subsequently processed with the LigPrep module of the Schrodinger suite 2017-1 to ensure proper conformational sampling [62]. The refinement processes, such as appropriate chiral position, protonation, and ionization state, were fixed using Epik at a target pH of 7.0 +/− 2.0. Furthermore, 32 possible stereoisomers were determined for each ligand. The entire process was carried out under the OPLS3 force field.

#### 4.4.2. Protein Preparation and Grid Generation

For docking studies, the proteins of interest were considered first. Therefore, the structure of AKT1, EGFR, and STAT3 (PDB ID: 3O96, 3W32, and 6NJS) was selected and downloaded from the RCSB protein databank and then subjected to a multistep process for structural correction using the protein preparation wizard from Schrodinger 2017-1 suite [30]. The structure was refined by adding the appropriate H-atoms, charges, and bond orders. Unnecessary water molecules were removed from ligand binding sites no longer than 5 Å. Missing atoms were filled and, subsequently, the protein structure was optimized using PROPKA and minimized using OPLS3 forcefield to restrain heavy atoms to RMSD 0.3 Å [63]. For docking simulations, the ligand binding site of the protein was determined by building a grid box around the reference ligand. During final grid generation, default settings were maintained, where the van der Waals scaling factor and partial charge cutoff were set at 1.00 and 0.25, respectively.

#### 4.4.3. Glide XP Docking

Glide extra-precision (XP) docking is considered a precise scoring function, and was employed in this study. XP docking is more extensive than standard-precision docking [64,65]. The process of XP docking was carried out in the Schrodinger 2017-1 suite, keeping the van der Waals scaling factor and partial charge cutoff at 0.80 and 0.15, respectively. After that, the complex with the lowest glide score was selected and considered for molecular dynamics simulation. 

### 4.5. MM-GBSA Binding Energy Calculation

All selected docking poses were used to calculate the binding free energy using the prime module from the Schrödinger suite, in which the force field was OPLS3 [63]. The MM-GBSA scores were calculated by introducing a Glide XP pose viewer file. The sum of the binding free energies was calculated as follows:ΔG_binding_ = G_docking complex_ − (G_protein_ + G_ligand_),
where G = EMM + GSGB + GNP.

In prime MM-GBSA, the binding free energy is the calculation of EMM, GSGB, and GNP. EMM denotes the molecular mechanics energies. GSGB is a polar solvation model, while GNP is characterized as non-polar solvation. The highest negative value of ΔG_bind_ is considered to indicate excellent binding affinity.

### 4.6. MD Simulation

The dynamic behavior of the target protein and its structural stability, conformational changes, and intermolecular interactions after ligand binding were verified by MD simulations conducted using the Desmond academic version (2020-1). The docked genistein-AKT1 complex was placed in an orthorhombic simulation box and solvated using a TIP3P solvation/aqueous system. The minimal distance between the protein and the box boundary was set to 10 Å. Further, an appropriate number of counter ions, such as Na^+^ and Cl^−^, were added to the system for neutralization, and the concentration was maintained at 0.15 M. The predefined system relaxation protocol in Desmond was followed by a total of eight stages, from equilibrating the system to the final simulation. NVT followed the second and third stages. In the second stage, the system used Brownian dynamics at 10 K for 100 ps to restrict the motion of the heavy atom of the protein, whereas the third stage followed the restriction on large atoms for 12 ps. The fourth stage maintained the NPT (constant temperature, constant pressure) condition, again with restraints on the solute-heavy atoms for 12 ps, and then solvated the pocket at stage five. Stage six was then followed by the NPT condition instead of NVT for 12 ps, with restraints on solute-heavy atoms, but stage seven involved a 24 ps simulation time without restraints under NPT conditions. The final simulation was performed for 200 ns following the NPT ensemble at a temperature of 300 K and a pressure of 1 bar. Each trajectory was saved at 25 ps intervals for the final analysis. Each complex was simulated three times with different random seed numbers.

## 5. Conclusions

Metastasis is a significant hallmark of lung cancer-related deaths worldwide. Therefore, we identified the top hub genes and determined whether they are related to metastasis modulators. Overall, our findings indicated that several target genes are involved in metastasis, of which *AKT1* directly promotes the progression of NSCLC through metastasis. Next, we unveiled the possibility of genistein binding to AKT1 through MD simulations, where genistein showed significant results compared with known reference compounds. However, wet laboratory experiments are required to validate these findings. 

## Figures and Tables

**Figure 1 molecules-29-01913-f001:**
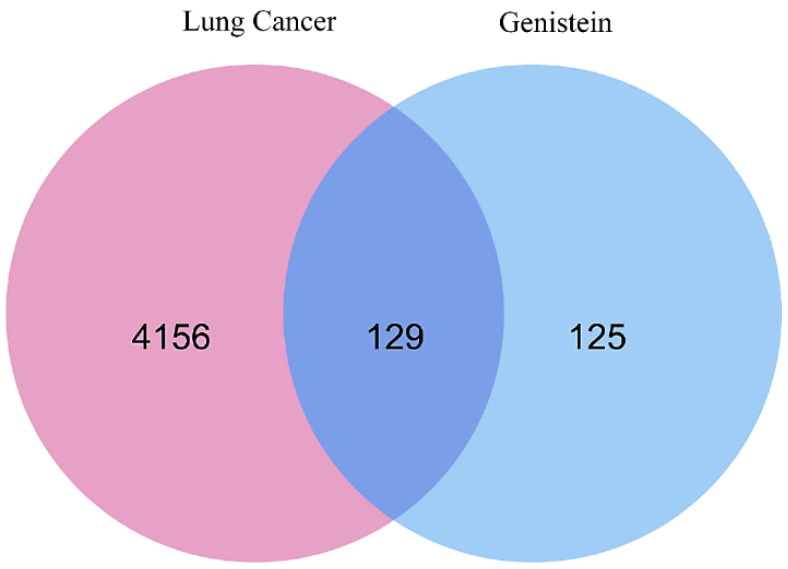
Graphical representation of overlapping gene symbols between lung cancer and genistein. A total of 129 overlapping genes are correlated and represented in blue color.

**Figure 2 molecules-29-01913-f002:**
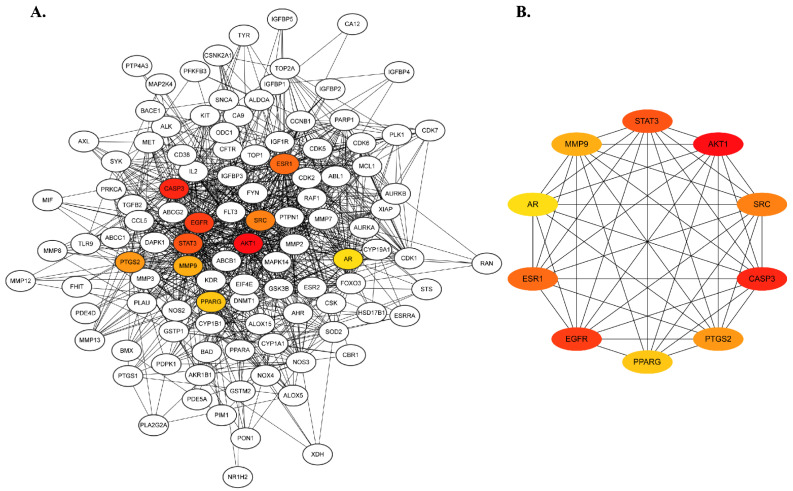
Protein–protein interaction network of (**A**) 129 overlapping genes obtained from the STRING database and (**B**) the top 10 hub genes based on degree centrality score. The hub genes are colored yellow to dark red, indicating the more significant role in the network.

**Figure 3 molecules-29-01913-f003:**
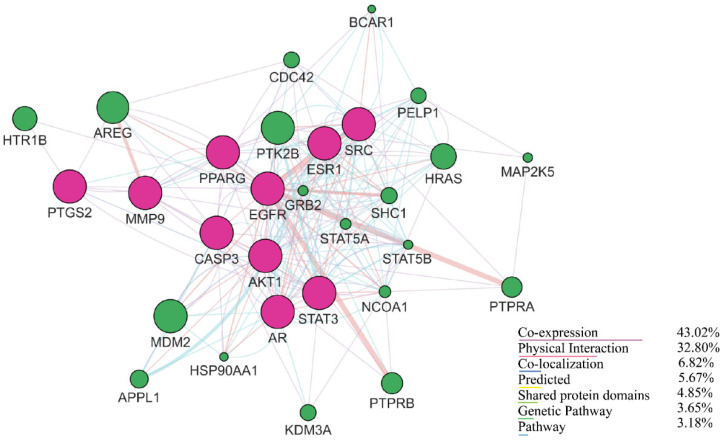
Gene–gene interaction network obtained using GeneMANIA, representing co-expression, physical interaction, colocalization, shared protein domains, and genetic pathways of hub genes. The hub genes are colored pink, and similar related genes are colored green.

**Figure 4 molecules-29-01913-f004:**
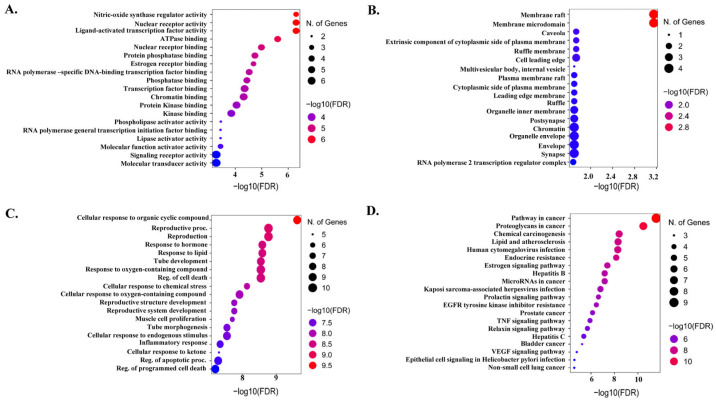
Gene ontology and Kyoto Encyclopedia of Genes and Genomes (KEGG) enrichment analysis. (**A**) Molecular function, (**B**) cellular component, (**C**) biological process, and (**D**) KEGG pathways.

**Figure 5 molecules-29-01913-f005:**
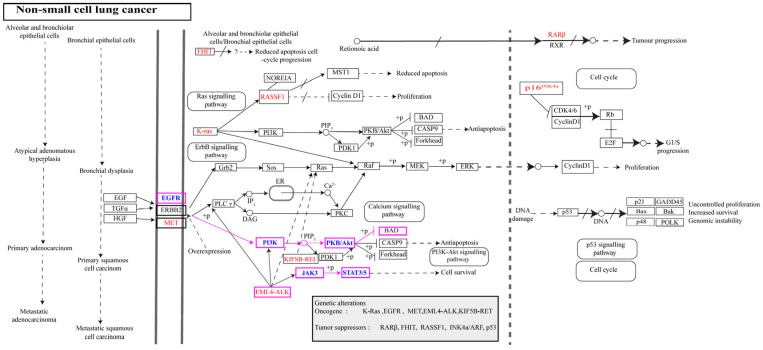
Non-small cell lung cancer KEGG pathway. The boxes marked in pink represent the key functional targets of genistein, and the arrows denote the transduction pathways involved in several cellular events.

**Figure 6 molecules-29-01913-f006:**
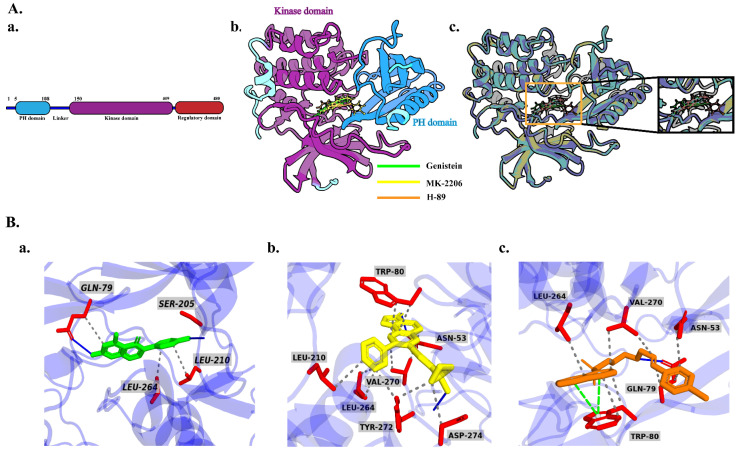
Overview of AKT1 structure and ligand binding site. (**A**) (**a**) Graphical representation of AKT1 isoform consisting of a pleckstrin homology (PH) domain, linker, kinase domain, and regulatory domain, (**b**) molecular docking of genistein, MK-2206, and H-89 with AKT1, and (**c**) superimposed three docked complexes. (**B**) Protein–ligand three-dimensional interactions of (**a**) AKT1-genistein, (**b**) AKT1-MK-2206, and (**c**) AKT1-H-89.

**Figure 7 molecules-29-01913-f007:**
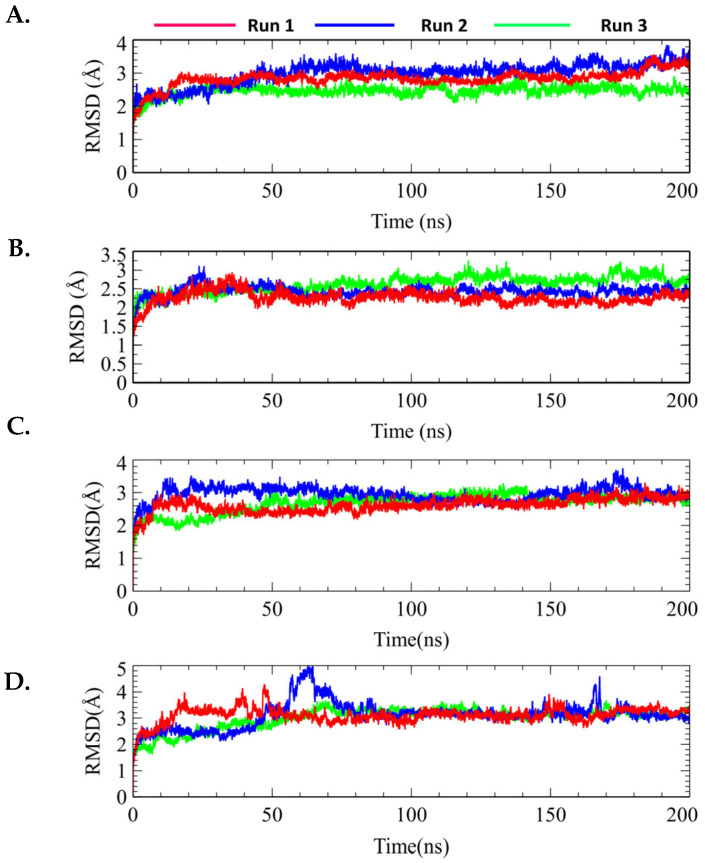
Protein root mean square deviation (RMSD) values of individual three simulation runs (run 1, run 2, and run 3) of (**A**) apo, (**B**) genistein-bound-AKT1, (**C**) MK-2206-bound-AKT1, and (**D**) H-89-bound-AKT1.

**Figure 8 molecules-29-01913-f008:**
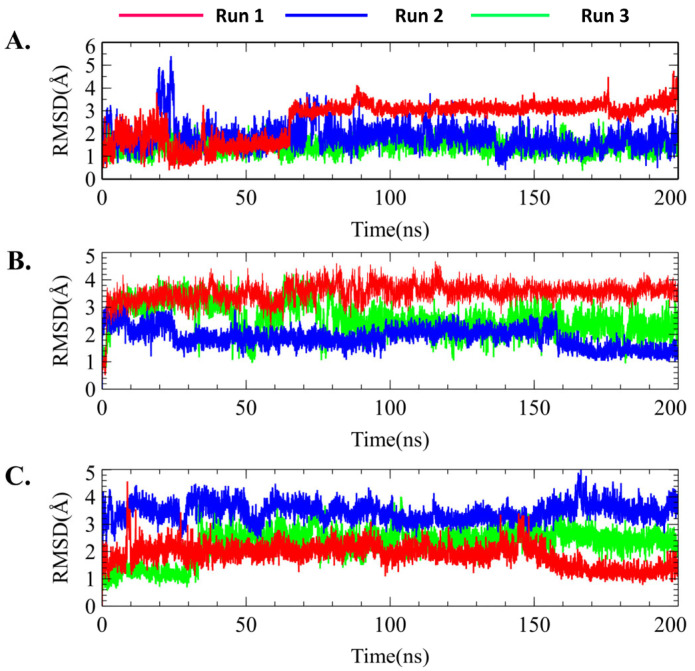
Ligand root mean square deviation (RMSD) values of individual three simulation runs (run 1, run 2, and run 3) of (**A**) genistein, (**B**) MK-2206, and (**C**) H-89.

**Figure 9 molecules-29-01913-f009:**
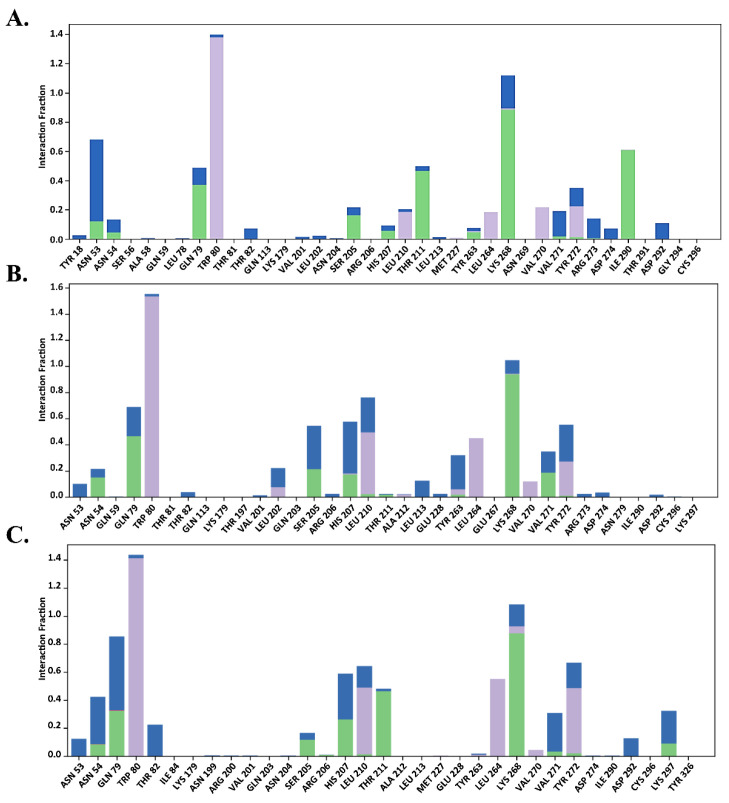
Total protein–ligand contact analysis during 200 ns of molecular dynamics simulation of genistein in (**A**) run 1, (**B**) run 2, (**C**) run 3. Hydrogen bonds, hydrophobic interactions, and water bridges are represented by green, light purple, and blue color.

**Figure 10 molecules-29-01913-f010:**
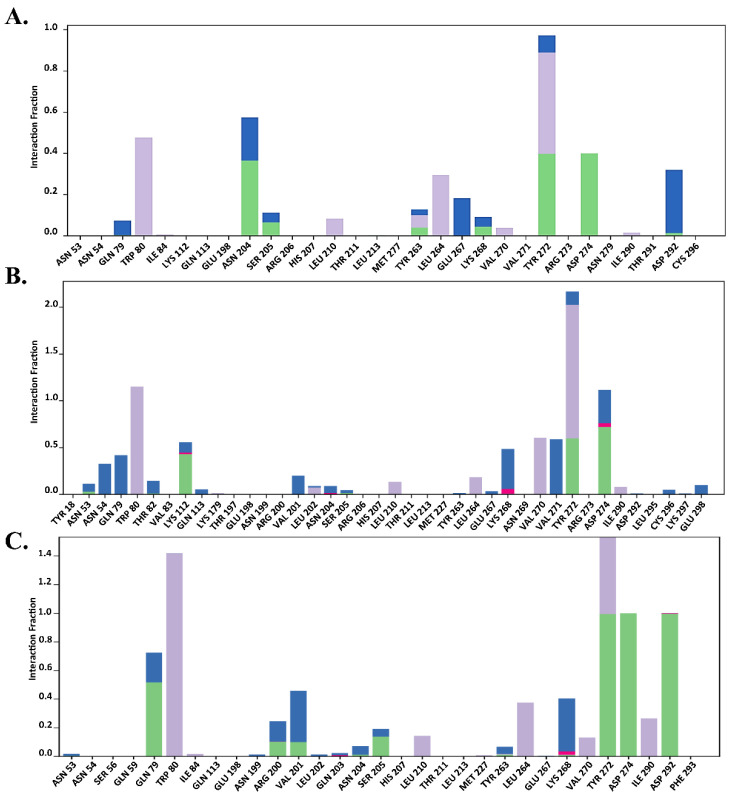
Total protein–ligand contact analysis during 200 ns of molecular dynamics simulation of MK-2206 in (**A**) run 1, (**B**) run 2, (**C**) run 3. Hydrogen bonds, hydrophobic interactions, ionic and water bridges are represented by green, light purple, red and blue color.

**Figure 11 molecules-29-01913-f011:**
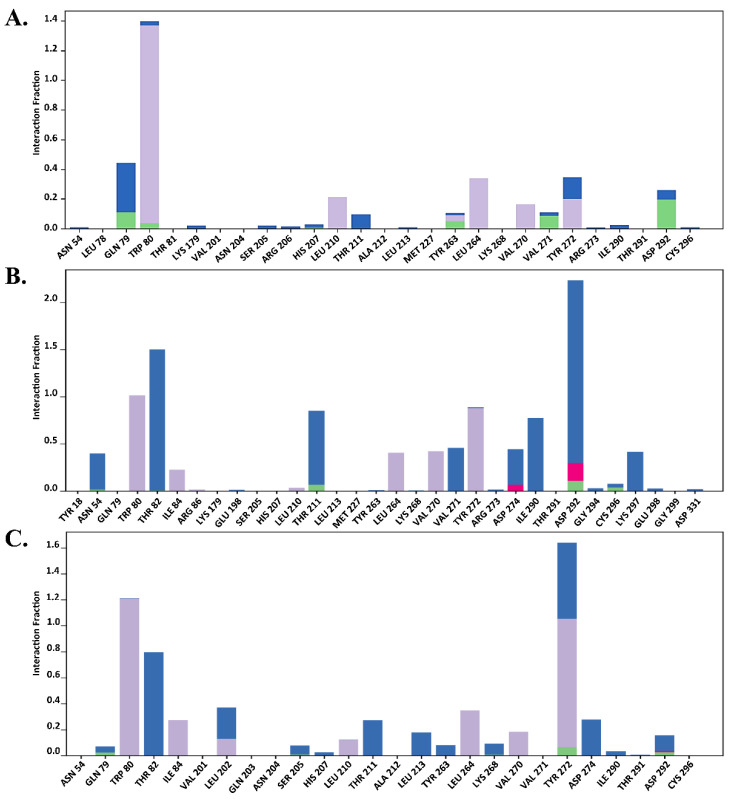
Total protein–ligand contact analysis during 200 ns of molecular dynamics simulation of H-89 in (**A**) run 1, (**B**) run 2, (**C**) run 3. Hydrogen bonds, hydrophobic interactions, ionic and water bridges are represented by green, light purple, red and blue color.

**Figure 12 molecules-29-01913-f012:**
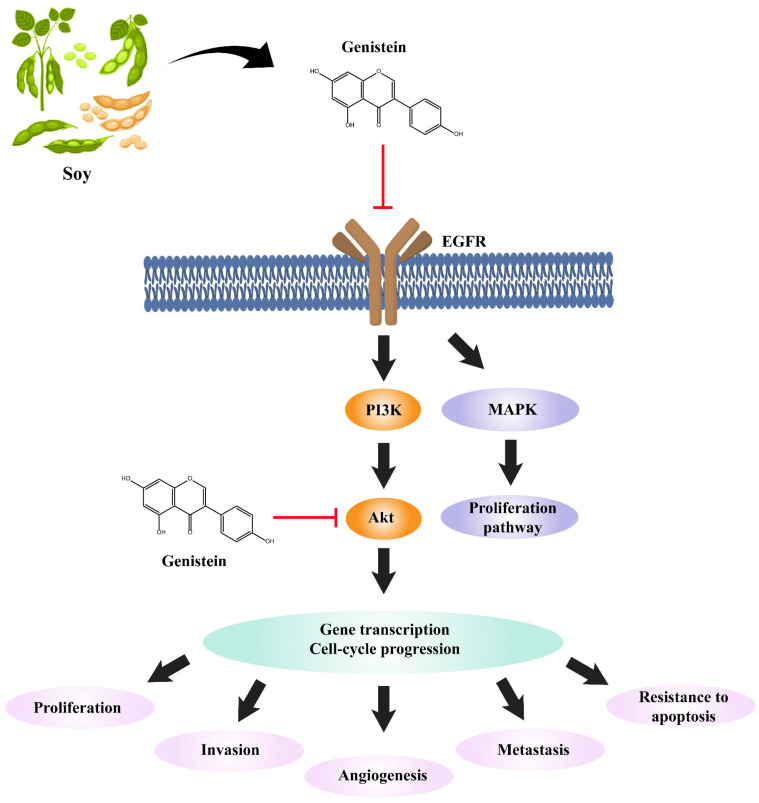
Graphical representation of the anticancer pathway modulated by genistein.

## Data Availability

The data presented in this study are available in article and Appendix A.

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
