# Peer review of "Identifying the Multitarget Pharmacological Mechanism of Action of Genistein on Lung Cancer by Integrating Network Pharmacology and Molecular Dynamic Simulation"

_molecules, 2024, doi:10.3390/molecules29091913_

Round 1
Reviewer 1 Report
Comments and Suggestions for Authors
Overall it seems to be a good review article by the authors. They have covered majority of biochemical aspects of the Genistein and lung cancer. Language is appropriate.
Author Response
Comments: Overall it seems to be a good review article by the authors. They have covered majority of biochemical aspects of the Genistein and lung cancer. Language is appropriate.
Answer: The authors are thankful for the appreciation of current research.
Reviewer 2 Report
Comments and Suggestions for Authors
The submitted manuscript explores the potential pharmacological role of genistein in inhibiting cancer-related proteins , employing a robust methodology. I have the following comments:
1. For a clearer comparison of the binding poses between genistein and two other compounds within AKT, it would be beneficial to align protein and visualize from te same prospect in Figure 6.
2. It is mentioned that MK-2206-bound AKT1 displays a slightly higher RMSD when compared to genistein-bound AKT1. What are the conformational changes associated with the elevated RMSD? Additionally, how does this relate to MK-2206 binding?
3. Figure 9 shows interaction fraction, which I assume represents the percentage of time an interaction is established. Could you clarify the cases when this fraction exceeds 100%?
Author Response
- For a clearer comparison of the binding poses between genistein and two other compounds within AKT, it would be beneficial to align protein and visualize from te same prospect in Figure 6.
Answer: Figure 6 has been updated. Figure 6Ac represents the superimposed all-docked complex.
- It is mentioned that MK-2206-bound AKT1 displays a slightly higher RMSD when compared to genistein-bound AKT1. What are the conformational changes associated with the elevated RMSD? Additionally, how does this relate to MK-2206 binding?
Answer: We updated the writings after multiple simulation times. The authors recommend reading the updated version.
- Figure 9 shows interaction fraction, which I assume represents the percentage of time an interaction is established. Could you clarify the cases when this fraction exceeds 100%?
Answer: Figure 9 summarizes the total protein-ligand contacts with specific subtypes of interaction, such as hydrogen bonds, hydrophobic interaction, ionic and water bridges contacts. In each figure, the x-axis represents the residue of the target protein, and the y-axis represents the interaction fraction. The stacked bar charts are normalized, where values in the y-axis suggest the percentage of simulation time, such as 0.3 or 0.4, which means 30% and 40% simulation time, respectively. Values over 1.0 (equivalent to 100%) represent more than one contact of the same subtype with particular ligands. We also updated the figure and legends of Figure 9.
Reviewer 3 Report
Comments and Suggestions for Authors
To authors,
This work by Das et al., showcases the application of network pharmacology in identifying key proteins in involved in NSCLC and its progression – AKT1, CASP3, EGFR, STAT3, 19 ESR1, SRC, PTGS2, MMP9, PRAG, and AR. They followed it up by computational characterization of genistein binding to EGFR, AKT1, and STAT3. While there is nothing new about the role of these proteins in NSCLC, understanding how genistein interacts with its molecular targets would provide important insights into its anti-apoptotic effects on tumor cell lines. The network pharmacology and genomic analysis were found to be sound and well performed. However, the molecular docking and dynamics simulations warrant a major do over in order that the conclusions drawn are aptly supported by the observations. Here are my major and minor concerns pertaining to the manuscript.
Major:
1. Molecular docking:
a) Results
The molecular docking analysis (Page 6) needs to be rewritten to highlight missing information. For example, the authors mention redocking was performed to avoid false docking poses and an accurate docking score. As a matter of fact, redocking is mainly used to ascertain if the docking protocol is able to predict the correct binding pose that is already known from a preexisting complex structure. However, it does not help in getting an accurate docking score, since docking score is an approximate value and is widely known to have a standard error of > 2.5 kcal/mol when compared to actual experimental binding affinity. The redocking statement must be rewritten to highlight this. Moreover, it is unclear which ligands were redocked to which proteins and the results associated with the redocking have not been mentioned anywhere in the main text or the supporting information. In the same section (Line 175-176), it has been written that “A docking score less than 2 Å supports the entire docking methodology”. This statement is entirely wrong. First, the docking score is a predicted binding affinity with units of kcal/mol and not angstrom as mentioned in the manuscript. Second, are the authors trying to imply if the RMSD between the redocked pose and the reference structure is below 2 Å, then the redocking process can be deemed as a success. A good docking program like Glide XP has a much stricter criterion of 1Å for redocking. The Line 175-176 need correction to highlight this fact.
The most glaring error in the results section is that there is no clarity of how poses for the said compounds were selected. First, how much ligand conformations were generated for each ligand in LigPrep? How much poses were generated for each input ligand conformation. Upon conclusion of docking, was the top ranked pose considered as the best pose and used for MM-GBSA analysis. If so, please clarify. As a standard practice, it is recommended to perform a rescoring of the docking results with independent scoring function like SMINA, DOCK, or DrugScoreX to get an unbiased pose selection. In the current case, no such procedure was performed, and the authors are strongly recommended to perform rescoring and update the results. Moreover, looking at the table S1, although genistein has a better docking score than MK-2206 or H-89 when docked to AKT1, the more rigorous and comparatively accurate MM-GBSA suggests that MK-2206 is better than genistein. The docking scores for genistein and MK-2206 are also within 0.5 kcal/mol of each other. Can the authors explain the rationale behind choosing genistein over MK-2206.
b) Materials and methods
The information of ligands used as input for LigPrep is missing along with the number of output conformations per input ligand. Also missing is how was the stereoisomers handling performed. Furthermore, in terms of protein preparation, regarding the proteins used as input, the PDB 3O96 corresponding to AKT1 has 76 missing residues, 3W32 has 13 missing residues, and 6NJ6 has 39 missing residues. Can the authors provide information how these missing residues were accounted for in docking. If not, it is strongly recommended to add these missing residues to the respective structures before the docking to get meaningful results, given the fact that the current work focuses on the allosteric pockets of the proteins, especially AKT1. Finally, for the grid selection process, it is unclear how the grid/docking region was defined. Was it centered on the center of the mass of the active site residue/s or on the co-crystallized ligand? Please provide a clear workflow to ensure reproducibility and to draw meaningful conclusions. In the current state, the section was found to be missing crucial information and needs updation.
2. Molecular dynamics:
a. Results
In the results section, the authors compare genistein, MK-2206, and H-89 bound to the allosteric pocket of AKT1. Here, RMSF analysis was missing that could highlight the effect of ligand binding on the residues of the AKT1 allosteric pocket in addition to RMSD providing a clear picture. Looking at the complex RMSD, both genistein and MK-2206 are equivalent with no noticeable differences in protein RMSD. If one considers bound ligand RMSD, it clear that genistein undergoes a conformational change around 65 and 160 ns while increasing even further at the end of 200 ns, while MK-2206 undergoes a change early (5 ns) and remains fairly stable till end of the 200 ns simulation. Looking at the MM-GBSA and ligand RMSD values, it appears that MK-2206 is a much better binder that genistein. Can the authors provide a rationale why genistein a better binder than MK-2206. Also, the authors have not provided standard deviations for the RMSD values that gives a much better idea of the overall fluctuations in the values for each of the complexes. Moreover, a glaring flaw in this study, is that all observations have been drawn from solitary 200ns simulations. The authors are recommended to perform replicate simulations of the same poses (total 3 for each system) which is standard in experimental techniques for a robust representation of observations and to draw meaningful conclusions.
b. Materials and methods
The MD simulation section gives an impression that only genistein bound to AKT1 was prepared and simulated while the results section shows that AKT1 bound to Mk-2206 and H-89 were also simulated. Please correct this discrepancy. Also, citation for desmond is missing and so is the description of post-simulation analysis. For e.g. clearly mention which atom selections were used to calculate protein and ligand rmsd, what was the reference structure used.
Minor:
1. Figure 6b can be color coded and labelled to highlight the important domains mentioned by the authors in the manuscript.
2. It would be better to depict the structures of the ligands including genistein in figure 6 or SI to better reflect the dataset being studied with docking.
3. Consistent numbering – In table S1, please follow a consistent decimal place for the docking scores as well as the GBSA values.
4. Inconsistent formatting of references – inconsistent formatting was observed in references with journal names sometimes appearing in small caps and large caps arbitrarily. Please stick to a consistent representation of the journal name. For example, please compare refs. 1, 2, 5, 14, 18, etc.
Comments on the Quality of English LanguageEnglish language fine. Minor spell check and formatting needed
Author Response
The authors are thankful for the your advice and reviewing.
Major:
1.Molecular docking:
a)Results
The molecular docking analysis (Page 6) needs to be rewritten to highlight missing information. For example, the authors mention redocking was performed to avoid false docking poses and an accurate docking score. As a matter of fact, redocking is mainly used to ascertain if the docking protocol is able to predict the correct binding pose that is already known from a preexisting complex structure. However, it does not help in getting an accurate docking score, since docking score is an approximate value and is widely known to have a standard error of > 2.5 kcal/mol when compared to actual experimental binding affinity. The redocking statement must be rewritten to highlight this. Moreover, it is unclear which ligands were redocked to which proteins and the results associated with the redocking have not been mentioned anywhere in the main text or the supporting information. In the same section (Line 175-176), it has been written that “A docking score less than 2 Å supports the entire docking methodology”. This statement is entirely wrong. First, the docking score is a predicted binding affinity with units of kcal/mol and not angstrom as mentioned in the manuscript. Second, are the authors trying to imply if the RMSD between the redocked pose and the reference structure is below 2 Å, then the redocking process can be deemed as a success. A good docking program like Glide XP has a much stricter criterion of 1Å for redocking. The Line 175-176 need correction to highlight this fact.
The most glaring error in the results section is that there is no clarity of how poses for the said compounds were selected. First, how much ligand conformations were generated for each ligand in LigPrep? How much poses were generated for each input ligand conformation. Upon conclusion of docking, was the top ranked pose considered as the best pose and used for MM-GBSA analysis. If so, please clarify. As a standard practice, it is recommended to perform a rescoring of the docking results with independent scoring function like SMINA, DOCK, or DrugScoreX to get an unbiased pose selection. In the current case, no such procedure was performed, and the authors are strongly recommended to perform rescoring and update the results. Moreover, looking at the table S1, although genistein has a better docking score than MK-2206 or H-89 when docked to AKT1, the more rigorous and comparatively accurate MM-GBSA suggests that MK-2206 is better than genistein. The docking scores for genistein and MK-2206 are also within 0.5 kcal/mol of each other. Can the authors explain the rationale behind choosing genistein over MK-2206.
Answer: The authors express their gratitude to the reviewer for his insightful comment and for his time in reviewing this work. The authors did the redocking to check the RMSD between the redocked pose and the reference ligand-bound complex. The ligand that was bound to AKT1 in 3O96 ID was used as a reference for redocking. The redocking output represented the RMSD, which was 0.030Ȧ. After that, docking was carried out between AKT1 and genistein, MK-2206, and H-89. The docking result was compared with genistein by selecting two known inhibitors. Our writing on docking methods has been updated to clarify each point.
It's inaccurate to say that any compound is superior to control solely on docking and MM-GBSA score, even though it has shown higher scores of docking and MM-GBSA than control. According to the authors, MK-2206 has a higher MM-GBSA than genistein. The authors conducted molecular dynamics simulations to clarify this point. Genistein achieved almost identical results with MK-2206 after run 1. The authors highly appreciate the reviewer's suggestion to execute multiple simulations. After performing three 200ns simulations (each complex) with random seed numbers, we obtained nearly identical results according to protein RMSD, ligand RMSD, total protein-ligand contacts, and timeline plot. We made changes to our writing. We recommend that authors read updated writing.
- b) Materials and methods
The information of ligands used as input for LigPrep is missing along with the number of output conformations per input ligand. Also missing is how was the stereoisomers handling performed. Furthermore, in terms of protein preparation, regarding the proteins used as input, the PDB 3O96 corresponding to AKT1 has 76 missing residues, 3W32 has 13 missing residues, and 6NJ6 has 39 missing residues. Can the authors provide information how these missing residues were accounted for in docking. If not, it is strongly recommended to add these missing residues to the respective structures before the docking to get meaningful results, given the fact that the current work focuses on the allosteric pockets of the proteins, especially AKT1. Finally, for the grid selection process, it is unclear how the grid/docking region was defined. Was it centered on the center of the mass of the active site residue/s or on the co-crystallized ligand? Please provide a clear workflow to ensure reproducibility and to draw meaningful conclusions. In the current state, the section was found to be missing crucial information and needs updation.
Answer: The authors suggest reading the updated version of the writings to check the method section about Ligprep and grid generation. Before docking, the authors checked to see if there was any missing residue in the 3O96 pdb. The authors focused on the allosteric site and verified no missing residue in the area.
2.Molecular dynamics:
- Results
In the results section, the authors compare genistein, MK-2206, and H-89 bound to the allosteric pocket of AKT1. Here, RMSF analysis was missing that could highlight the effect of ligand binding on the residues of the AKT1 allosteric pocket in addition to RMSD providing a clear picture. Looking at the complex RMSD, both genistein and MK-2206 are equivalent with no noticeable differences in protein RMSD. If one considers bound ligand RMSD, it clear that genistein undergoes a conformational change around 65 and 160 ns while increasing even further at the end of 200 ns, while MK-2206 undergoes a change early (5 ns) and remains fairly stable till end of the 200 ns simulation. Looking at the MM-GBSA and ligand RMSD values, it appears that MK-2206 is a much better binder that genistein. Can the authors provide a rationale why genistein a better binder than MK-2206. Also, the authors have not provided standard deviations for the RMSD values that gives a much better idea of the overall fluctuations in the values for each of the complexes. Moreover, a glaring flaw in this study, is that all observations have been drawn from solitary 200ns simulations. The authors are recommended to perform replicate simulations of the same poses (total 3 for each system) which is standard in experimental techniques for a robust representation of observations and to draw meaningful conclusions.
Answer: We updated our writing after multiple simulation times.
b.Materials and methods
The MD simulation section gives an impression that only genistein bound to AKT1 was prepared and simulated while the results section shows that AKT1 bound to Mk-2206 and H-89 were also simulated. Please correct this discrepancy. Also, citation for desmond is missing and so is the description of post-simulation analysis. For e.g. clearly mention which atom selections were used to calculate protein and ligand rmsd, what was the reference structure used.
Answer: We updated our writing after multiple simulation times.
Minor:
1.Figure 6b can be color coded and labelled to highlight the important domains mentioned by the authors in the manuscript.
Answer: Figure 6 has been updated. According to the reviewer’s suggestion, figure 6Ab represents the newly updated figure.
2.It would be better to depict the structures of the ligands including genistein in figure 6 or SI to better reflect the dataset being studied with docking.
Answer: Figure 6 has been updated.
3.Consistent numbering – In table S1, please follow a consistent decimal place for the docking scores as well as the GBSA values.
Answer: The supplementary file has been updated.
4.Inconsistent formatting of references – inconsistent formatting was observed in references with journal names sometimes appearing in small caps and large caps arbitrarily. Please stick to a consistent representation of the journal name. For example, please compare refs. 1, 2, 5, 14, 18, etc.
Answer: I modified everything according to the MDPI reference format.
Reviewer 4 Report
Comments and Suggestions for Authors
The authors have done excellent work in identifying target genes against lung cancer, a disease that affects millions of people around the world. To achieve a better understanding of the manuscript it is suggested:
1- When identifying genes, why did the authors take a relevance score ≥ 20? Could you explain this in the results?
2- Increase the size of figure-2 so that it is clearer and more readable.
3- Make figures 4 and 5 clearer, it is not understood.
4- In Figure 7A a legend should be placed indicating the meaning of each color in the RMSD parameter.
5- In figure-9, what does the y axis of the Interaction Fraction mean? Please clarify the term and its meaning.
Author Response
First of all, This manuscript was already edited quality of English through Editage. Certificate of english editing was uploaded. (Please see the attachment.)
Answers of your questions are here.
1- When identifying genes, why did the authors take a relevance score ≥ 20? Could you explain this in the results?
Answer: The relevance score is a reflection of the relationship between genes and lung cancer, with higher scores indicating a stronger connection.
2- Increase the size of figure-2 so that it is clearer and more readable.
Answer: We updated the figure 2 with legends.
3- Make figures 4 and 5 clearer, it is not understood.
Answer: We updated figures 4 and 5.
4-In Figure 7A a legend should be placed indicating the meaning of each color in the RMSD parameter.
Answer: We updated the figures 7 and 8.
5- In figure-9, what does the y axis of the Interaction Fraction mean? Please clarify the term and its meaning.
Answer: Figure 9 summarizes the total protein-ligand contacts with specific interaction subtypes, such as hydrogen bonds, hydrophobic interaction, ionic and water bridge contacts. In each figure, the x-axis represents the target protein's residue, and the y-axis represents the interaction fraction. The stacked bar charts are normalized, where values in y-axis suggest the percentage of simulation time, such as 0.3 or 0.4 means 30% and 40% simulation time, respectively. Values over 1.0 (equivalent to 100%) represent more than one contact of the same subtype with particular ligands. To clarify, we also updated the figure and legends of Figure 9.

Round 2
Reviewer 3 Report
Comments and Suggestions for Authors
Minor text editing needed.
1. L173-174 spacing between value and angstrom
2. L256 - protein-lignad must be protein-ligand.
3. L415-417 - unclear what the authors are referring to.
Author Response
Dear reviewer
Thank you for your comments of my manuscript.
Here is the answers of your comments.
1. L173-174 spacing between value and angstrom
Answer: We corrected the spacing between value and angstrom.
2. L256 - protein-ligand must be protein-ligand.
Answer: We corrected it to protein-ligand.
3. L415-417 - unclear what the authors are referring to.
Answer: We updated the writings.